# Protopine-Type Alkaloids Alleviate Lipopolysaccharide-Induced Intestinal Inflammation and Modulate the Gut Microbiota in Mice

**DOI:** 10.3390/ani14152273

**Published:** 2024-08-05

**Authors:** Jialu Huang, Meishan Yue, Yang Yang, Yisong Liu, Jianguo Zeng

**Affiliations:** 1College of Veterinary Medicine, Shanxi Agricultural University, Jinzhong 030801, China; hhjj8181@163.com (J.H.); 18404984883@163.com (M.Y.); 2College of Veterinary Medicine, Hunan Agricultural University, Changsha 410128, China; yangyang0616@foxmail.com (Y.Y.); liuyisong@hunau.edu.cn (Y.L.)

**Keywords:** MPTA, gut, inflammation, microbiota, mice

## Abstract

**Simple Summary:**

The study investigates the therapeutic effects of protopine type alkaloid (MPTA) derived from *Macleaya cordata* (Willd) R. Br. in a mouse model of intestinal inflammation induced by lipopolysaccharide (LPS). The research demonstrates MPTA’s ability to alleviate intestinal inflammation, increase intestinal goblet cell abundance and reduce pro-inflammatory cytokines. MPTA also modulates the gut microbiota and volatile short-chain fatty acids, suggesting its potential as a treatment for gastrointestinal diseases.

**Abstract:**

In this study, we assessed the therapeutic effects of *Macleaya cordata* (Willd). R. Br.-derived protopine-type alkaloids (MPTAs) in a mouse model of lipopolysaccharide (LPS)-induced intestinal inflammation. The experimental design involved the allocation of mice into distinct groups, including a control group, a model group treated with 6 mg/kg LPS, a berberine group treated with 50 mg/kg berberine hydrochloride and low-, medium- and high-dose MPTA groups treated with 6, 12 and 24 mg/kg MPTAs, respectively. Histological analysis of the ileum, jejunum and duodenum was performed using Hematoxylin and Eosin (H&E) staining. Moreover, the quantification of intestinal goblet cells (GCs) was performed based on PAS staining. The serum levels of IL-1β, IL-6, IL-8 and TNF-α were quantified using an enzyme-linked immunosorbent assay (ELISA), while the mRNA levels of TLR4, NF-κB p65, NLRP3, IL-6 and IL-1β were assessed using quantitative PCR (qPCR). The protein levels of TLR4, Md-2, MyD88, NF-κB p65 and NLRP3 were determined using Western blotting. Furthermore, the 16S rDNA sequences of bacterial taxa were amplified and analysed to determine alterations in the gut microbiota of the mice following MPTA treatment. Different doses of MPTAs were found to elicit distinct therapeutic effects, leading to enhanced intestinal morphology and an increased abundance of intestinal GCs. A significant decrease was noted in the levels of pro-inflammatory cytokines (IL-1β, IL-6, IL-8 and TNF-α). Additionally, the protein levels of TLR4, MyD88, NLRP3 and p-p65/p65 were markedly reduced by MPTA treatment. Furthermore, 16S rDNA sequencing analysis revealed that the administration of 24 mg/kg MPTAs facilitated the restoration of microbial composition.

## 1. Introduction

Intestinal diseases caused by pathogens can damage intestinal epithelial cells. This damage leads to intestinal inflammation, which can cause diarrhoea in livestock and poultry [1]. It can also destroy the physical barrier of the intestinal mucosa and reduce enzyme activity, leading to electrolyte imbalance and the abnormal decomposition and absorption of nutrients [2,3]. The presence of Gram-negative bacteria poses a significant threat to public health, environmental stability and food safety [4]. Bacterial infection can lead to the rupture of the cell membrane, resulting in the release of lipopolysaccharide (LPS), a potent immunostimulant that plays a crucial role in promoting inflammation [5]. Although inflammation serves as a defence mechanism against infection, excessive inflammation can have detrimental effects and contribute to the development of diseases [6]. The induction of LPS triggers an inflammatory response and causes damage to intestinal cells, resulting in acute diarrhoea [7]. LPS can specifically stimulate Toll-like receptor 4 (TLR4) to produce pleiotropic cytokines/chemokines [8] and can prime NLRP3 activation [9]. Myeloid differentiation protein-2 (MD-2) is a glycoprotein that binds to LPS and TLR4 [10]. It plays a pivotal role in inflammation and is indispensable for the activation of TLR4 signalling [11]. Furthermore, LPS can elicit systemic inflammation, activate immune cells, trigger sepsis and induce GSDMD-mediated proptosis [12].

The intestine, an organ with the highest metabolic activity in the body, is highly susceptible to external influences [13]. Consequently, intestinal inflammation is often caused by an imbalance in the ecological equilibrium of the intestinal microbiota [14]. The intestinal microbiota plays a crucial role in host nutrient metabolism by providing energy and enzymes to support the host’s well-being [15]. Microbial by-products, such as short-chain fatty acids (SCFAs), can be absorbed and utilised by the host [16]. Disturbances in the intestinal flora and their metabolites are directly associated with the occurrence of diarrhoea [17]. Approximately 10–15% of energy in the animal body is derived from fermentation by-products of the intestinal flora (SCFAs). In particular, acetic acid and propionic acid contribute to energy metabolism in the liver, while butyric acid serves as a primary energy source for intestinal epithelial cells [18]. SCFAs can regulate various systems, such as the gut, nervous, endocrine and blood systems. Moreover, they play a role in different stages of inflammatory processes during diseases through immune cell regulation [19,20].

Numerous studies have demonstrated that plant extracts exert protective effects against diarrhoea by regulating ion transport or aquaporins, maintaining water and electrolyte balance, suppressing the inflammatory response, targeting gastrointestinal hormone secretion and protecting the intestinal mucosa [21,22,23]. *Macleaya cordata* (Willd.) R. Br., also known as *bóluòhúi*, is a plant belonging to the *Papaveraceae* family. It is native to China and is extensively utilised in animal husbandry across Asia, Europe and the Americas [24]. Numerous studies have showcased the effectiveness of *M. cordata* extract in alleviating intestinal inflammation, modulating the intestinal microbiota and enhancing livestock growth [25,26]. Additionally, *M. cordata* extract exhibits remarkable antibacterial properties and helps combat bacterial resistance [27]. *M. cordata* (Willd). R. Br.-derived protopine-type alkaloids (MPTAs) primarily consist of 35% protopine (PRO) and 15% allocryptopine (ALL) [28]. PRO can mitigate DSS-induced colitis by inhibiting inflammation, in addition to preserving the integrity of the intestinal mucosal barrier and modulating the composition of the gut microbiome [29,30]. ALL blocks the CX3CL1/GNB5/AKT2/NF-κB/apoptosis pathway and enhances intestinal barrier function to reduce colitis [31]. MPTAs exhibit antibacterial and anti-inflammatory properties, rendering them a suitable candidate for the treatment of diarrhoea; however, the precise mechanism underlying their efficacy against intestinal diseases remains elusive.

In the present study, a murine model of LPS-induced intestinal inflammation was used to determine the anti-inflammatory mechanism of MPTAs. In particular, the NF-κB and NLRP3 inflammasome signalling pathways were assessed. Moreover, changes in the intestinal microbiota and volatile SCFAs were evaluated. Our findings provide valuable insights into the potential therapeutic applications of MPTAs against animal gastrointestinal diseases, which pose a significant concern in animal healthcare.

## 2. Materials and Methods

### 2.1. Chemicals and Reagents

MPTAs (Lot No. 140501) were produced by Hunan MICOLTA Biological Resources Co., Ltd. (Changsha, China). MPTAs consist of 35% PRO and 15% ALL; the remaining components have been identified in prior research [28]. LPSs (*Escherichia coli* 055:B5) and lidocaine were obtained from Beijing Solarbio Science & Technology Co., Ltd. (Beijing, China). Berberine hydrochloride was purchased from Northeast Pharmaceutical Group Co., Ltd. (Shenyang, Liaoning, China). All other chemicals, solvents and reagents were of analytical grade.

### 2.2. Animals and Experimental Design

All animal experiments were reviewed and approved by the local committee of animal use and protection. Specific pathogen-free (SPF) male ICR mice were initially obtained from Hunan SJA Laboratory Animal Co., Ltd. (Changsha, China) The experiments were approved by the Ethics Committee of Hunan Agricultural University, China (No. 43321543).

The mice were maintained at the animal facility of Hunan Agricultural University. Sixty male ICR mice (3–4 weeks old, 20–25 g) were acclimatised for 1 week before the experiment. The animals were separated into six groups (*n* = 10): untreated mice (control group), mice administered only LPS (LPS group), mice administered LPS and berberine hydrochloride (50 mg/kg, LPS + Berberine group), mice administered LPS and a low dose of MPTA (6 mg/kg, LPS + L-MPTA group), mice administered LPS and a medium dose of MPTA (12 mg/kg, LPS + M-MPTA group) and mice administered LPS and a high dose of MPTA (24 mg/kg, LPS + H-MPTA group). The groups with LPS-induced diarrhoea were given a one-time intraperitoneal injection of LPS (6 mg/kg) on the eighth day [32]. Then, the mice were sacrificed, and their organs were harvested for analysis (Figure 1).

### 2.3. Sample Collection

Blood was collected and immediately centrifuged at 3500 rpm for 5 min at 4 °C and stored at −80 °C until analysis. The duodenum, jejunum, ileum and colon were removed and dissected into two parts: one portion was fixed with 4% paraformaldehyde for haematoxylin and eosin (H&E) staining and periodic acid–Schiff (PAS) staining, while the other portion was preserved for Western blotting and qPCR. The contents of all intestinal segments were stored in lyophilised tubes at −80 °C for 16S rDNA analysis and SCFA detection.

### 2.4. Determination of Diarrhoea Index (DI)

Each mouse is individually placed in a mouse cage that is equipped with a layer of filter paper, and the filter paper is replaced every hour. The DI of the mice is calculated and compared between groups within 5 h. The calculation of diarrhoea index refers to relevant reports: diarrhoea index = loose stool rate (the number of loose stools excreted by each mouse/total number of stools) × loose stool level (graded by the area of dirt formed by loose stool pollution filter paper, divided into 4 levels. Level 1: pollution diameter < 1 cm; Level 2: 1 cm ≤ pollution diameter < 2 cm; Level 3: 2 cm ≤ pollution diameter ≤ 3 cm; Level 4: pollution diameter > 3 cm), and the level of each pile of loose stools is separately counted (loose stool level = the total number of loose stool levels of a single mouse/the total number of loose stools). For the measurement of the diameter of the series, the diameter of circular faeces is directly measured, and the diameter of elliptical faeces is taken as the average of its longest and approximate circular diameters. The standard for distinguishing between dry and loose stool is based on whether there are stains on the filter paper. The statistics of faecal frequency were 1 count per particle or per pile (for those whose particle count cannot be distinguished) [33].

### 2.5. Western Blotting

In brief, the tissues were lysed using RIPA buffer (CW2333S, CWBio, Beijing, China) supplemented with protease and phosphatase inhibitors. The homogenates were centrifuged at 13,000 rpm at 4 °C for 20 min. The protein concentration in the supernatants was determined using enhanced BCA protein assay kits, following which the supernatants were preserved at −80 °C for future examination. After electrophoresis, proteins were transferred onto a polyvinylidene difluoride (PVDF) membrane (Invitrogen, Carlsbad, CA, USA) at 120 V. The following primary antibodies were used: TLR4 (D8L5W) rabbit mAb (#14358, 1:1000, CST, Boston, MA, USA), anti-MD2 antibody (ab24182, 1:1000, Abcam, Boston, MA, USA), MyD88 (D80F5) rabbit mAb (#4283, 1:1000, CST, Boston, MA, USA), NF-κB p65 monoclonal antibody (66535-1-Ig, 1:1000, Proteintech, Chicago, IL, USA), phospho-NF-κB p65 (Ser536) rabbit mAb (#3033, 1:1000, CST, Boston, MA, USA), anti-GSDMD antibody (ab219800, 1:1000, Abcam, Boston, MA, USA), anti-NLRP3 antibody (ab263899, 1:1000, Abcam, Boston, MA, USA), IL18 rabbit pAb (A1115, 1:1000, Abclonal, Wuhan, China) and β-actin rabbit mAb (high dilution) (AC026, 1:50,000, Abclonal, Wuhan, China). Appropriate secondary antibodies (1:5000, CWBio, Beijing, China) were added and incubated with the membrane for 1 h at room temperature. Then, the blot bands were visualised using BeyoECL Star (P0018AM, Beyotime, Shanghai, China). Imaging was performed using the ChemiDoc Imaging System (GeiView 6000 Pro, Bltlux, Guangzhou, China). The non-saturated bands were chosen for densitometric quantification using ImageJ software (ImageJ 1.51, Bethesda, MD, USA). Western blot analyses were repeated at least thrice.

### 2.6. Enzyme-Linked Immunosorbent Assay (ELISA)

Commercially available ELISA kits were used to measure the serum levels of IL-1β (EK201B/3, MultiSciences Biotech, Hangzhou, China), IL-6 (CSB-E04639m, Cusabio, Wuhan, China), IL-8 (EMC104, NeoBioscience, Shenzhen, China) and TNF-α (SEA133Mu, Cloud-Clone, Beijing, China), according to the manufacturers’ instructions. The absorbance value at 450 nm was measured. All samples were run in duplicates.

### 2.7. PAS Staining

The tissues fixed with paraformaldehyde were embedded, sectioned and stained with PAS reagent. Goblet cells (GCs) were then microscopically observed (50 μm, 400×). For each group, three fields of view were examined under an optical microscope, and the GC counts were manually recorded by two independent researchers.

### 2.8. H&E Staining 

For histological analysis, the tissues were fixed in 10% buffered formalin, dehydrated, embedded in paraffin, sectioned at 5 mm thickness and stained with H&E stain. A light microscope (Nikon, Tokyo, Japan) was used for histological analysis. ImageJ software was used to select and measure five relatively complete villi height and five crypts depth for each tissue section.

### 2.9. DNA Extraction and PCR Amplification

Total genomic DNA was extracted from 36 samples using the E.Z.N.A.^®^ Soil DNA Kit (Omega Bio-tek, Norcross, GA, USA) according to the manufacturer’s instructions. The DNA extract was analysed on agarose gel, and its concentration and purity were determined using a NanoDrop 2000 UV-vis spectrophotometer (Thermo Scientific, Wilmington, NC, USA). The V3–V4 hypervariable region of the bacterial 16S rRNA gene was amplified using the primer pair 338F (5′-ACTCCTACGGGAGGCAGCAG-3′) and 806R (5′-GGACTACHVGGGTWTCTAAT-3′), and PCR was performed in triplicate. The PCR product was extracted from agarose gel, purified and quantified using a Quantus™ fluorometer (Promega, Madison, WI, USA) [34].

### 2.10. Illumina MiSeq Sequencing and Processing of Sequencing Data

The purified amplicons were combined in equimolar ratios and subjected to paired-end sequencing on an Illumina MiSeq PE300 platform or a NovaSeq PE250 platform (Illumina, San Diego, CA, USA) following standard protocols established by Majorbio Bio-Pharm Technology Co., Ltd. (Shanghai, China). The resulting raw reads were archived in the NCBI Sequence Read Archive (SRA) database under the BioProject ID PRJNA24. Subsequently, the raw 16S rDNA gene sequencing reads were demultiplexed, quality filtered using fastp (version 0.20.0) and merged using FLASH (version 1.2.7). Operational taxonomic units (OTUs) were clustered at a 97% similarity cut-off by using UPARSE, and chimeric sequences were identified and removed. The taxonomy of each OTU representative sequence was analysed using the RDP Classifier (version 2.2) against the 16S rDNA database by considering a confidence threshold of 0.7. Taxonomic information was annotated for each representative sequence. The α-diversity indices, including Chao1, ACE, Shannon and Simpson indices, were calculated using QIIME (version 1.7.0). The β-diversity analysis involves principal coordinate analysis (PCoA) and non-metric multidimensional scaling (NMDS) ordination, which was performed using R software (version 2.15.3) based on weighted UniFrac distances. The Spearman rank correlation test was performed to analyse the relationships. The metabolic functions of the microbiota were predicted using Phylogenetic Investigation of Communities by Reconstruction of Unobserved States (PICRUSt V1.0.0). Linear discriminant analysis effect size (LEfSe) was employed to assess the differences between the groups. LEfSe first applies the Kruskal–Wallis rank-sum test to assess the differences in gene abundance between all groups and then uses the Wilcoxon rank-sum test to compare the pairs of groups. Finally, the significant differences were sorted based on linear discriminant analysis (LDA) scores [33].

### 2.11. Detection of Relative mRNA Expression by qPCR

The total RNA was extracted from the small intestinal tissue of the mice and transcribed into cDNA based on the manufacturer’s instructions. qPCR was performed on a real-time system using SYBR Green to detect the mRNA levels of TLR4, p65, NLRP3, and IL-1β. The primer sequences are listed in Table 1. The programme included pre-denaturation at 95 °C for 2 min, followed by 40 cycles at 95 °C for 15 s and 60 °C for 30 s. The GAPDH gene was used as an internal reference gene for normalisation. The experiments were performed in triplicate. The relative mRNA expression level was calculated using the 2^−ΔΔCt^ method.

### 2.12. Quantification of SCFAs

The intestinal sample was homogenised and centrifuged at 1000 rpm and 4 °C for 10 min. The supernatants were mixed with a 25% metaphosphoric acid solution in a 9:1 (*v*/*v*) ratio. After centrifugation and incubation at 4 °C for 30 min, the supernatants were transferred to chromatographic vials equipped with 1 µL inserts before performing gas chromatography–mass spectrometry (GC–MS). The analysis was performed on a Shimadzu GC-2010 Plus gas chromatograph (Shimadzu, Kyoto, Japan) with a DB-FFAP column (30 m × 250 μm × 0.25 μm).

The oven temperature was held at 60 °C for 3 min and then increased to 220 °C at 20 °C/min and maintained for 1 min. Data acquisition and processing were performed using LabSolutions GC–MS software (version 4.11, Shimadzu, Kyoto, Japan). The identified peaks were compared with retention time and mass spectral fragmentation data of standards. For the quantitative analysis of volatile compounds, the analyte concentration was calculated as the ratio of the peak area of the compound to the peak area of an internal standard (IS) [35].

### 2.13. Statistical Analyses

The results are presented as the mean ± standard error of the mean (SEM). The data were analysed using one-way analysis of variance (ANOVA). A one-tailed, unpaired Student’s *t*-test was performed for pairwise comparisons between two groups. For comparisons involving more than two groups, Tukey’s multiple-comparison test was employed. Statistical analyses were performed using GraphPad Prism (version 8.0, La Jolla, CA, USA). Differences were considered significant at a *p*-value of <0.05 (*) or <0.01 (**).

## 3. Results and Discussion

### 3.1. Effects of MPTAs on Body Weight of LPS-Induced Mice

Weight change is an important indicator of intestinal inflammation; this indicator can be used to develop prevention and treatment options for inflammatory lesions in animal models [36]. The body weight of mice in each group is shown in Figure 2A. Before intraperitoneal LPS injection, no significant difference was noted in the weight of mice in each group (*p* > 0.05), regardless of the doses of MPTAs administered. The final weight of mice in the LPS + M-MPTA group was significantly higher (103%) than the initial weight. In contrast, the body weight of mice in the LPS group reduced to 92% of their initial weight. As illustrated in Figure 2B, the weight change in mice was significantly higher in the LPS group than in the control group (*p* < 0.01). However, the mice administered various doses of MPTAs exhibited significantly lesser weight loss than those in the LPS group (*p* < 0.05). Notably, the weight change in the group administered a medium dose (12 mg/kg) of MPTAs was similar to that in the group administered berberine hydrochloride, which can effectively ameliorate weight loss (*p* < 0.01).

### 3.2. Effects of MPTAs on Mental State, Coats and Diarrhoea Symptoms of LPS-Induced Mice

The mice in the control group exhibited typical coats, activity levels, food intakes and defecation patterns. On treatment with LPS, the mental state of the mice declined after 2 h; however, diarrhoea was not observed. By 6 h, the mice in the LPS group began to exhibit signs of illness, including dishevelled coats, varying degrees of diarrhoea (characterised by yellow, loose stools) and Intestinal whitening, while those in the control group did not (Figure 2C). Throughout the experiment, the mice in the LPS group consistently displayed various symptoms, such as diarrhoea, the presence of blood in the stool, decreased appetite and dishevelled coats. In contrast, the mice in the LPS + Berberine group and the MPTA groups performed well and gradually regained their vitality on drug treatment. The stool consistency of these mice was recovered, and the amount of blood in their stool was reduced.

### 3.3. Effects of MPTAs on Diarrhoea Symptoms

As shown in Figure 2D, clinical symptoms of diarrhoea were noted in the mice before and after MPTA administration. Compared with the LPS group, the mice in the control group exhibited lumpy intestinal contents, a clean anus and granular faeces. However, the faeces of the LPS group mice had light yellow, purulent mucus, and their contents were yellow and thin. The DI of mice in the LPS group was 1.92. The DI of mice receiving a continuous intragastric administration of 24 mg/kg MPTAs was 1.41; this value was only 1.52 in the LPS + Berberine group, indicating that MPTAs exhibit a protective effect in mice with diarrhoea.

### 3.4. Effects of MPTAs on Intestinal Morphology of LPS-Induced Mice

As shown in Figure 3A and Appendix A, the walls of the small intestinal tissues (duodenum, jejunum and ileum) were intact in the control group mice. Their structures were clear, and villi were neatly arranged. Moreover, there was an obvious crypt structure and no evidence of inflammatory cell infiltration. In contrast, the intestinal walls of the mice in the LPS group appeared destroyed and necrotic. In these mice, the small intestinal villi had fallen off, the crypt structure had completely disappeared, and several inflammatory cells were present. In the LPS + Berberine, LPS + M-MPTA and LPS + H-MPTA groups, the small intestinal tissue of the mice returned to normal, and the intestinal walls appeared healthy. Moreover, the crypts appeared normal and inflammatory cells disappeared, indicating the beneficial effects of MPTA treatment.

The villus height (VH) and crypt depth (CD) of the small intestine (duodenum, jejunum and ileum) represent, to a certain extent, the shape of the small intestine and its digestion and absorption capacities. The shortening of villi and reduction in the VH/CD ratio indicate the presence of inflammation in the intestinal mucosa [37]. As shown in Figure 3B, the VH of the ileum, jejunum and duodenum in the LPS group was 66%, 87% and 61% of that in the control group, respectively, while the VH/CD ratio was 57%, 76% and 47% of that in the control group, respectively. In all cases, the difference was statistically significant (*p* < 0.01), suggesting that the intraperitoneal administration of LPS effectively induces intestinal inflammation in mice. Compared with the LPS group, treatment with L-MPTAs significantly increased the VH of the duodenal segment (*p* < 0.01), while also increasing the VH/CD ratio (*p* < 0.01). The VH of the duodenal segment in the LPS + H-MPTA group exhibited an extremely significant increase (*p* < 0.01); the VH/CD ratio also increased significantly (*p* < 0.01). Overall, these findings demonstrate that MPTAs can improve the intestinal morphology of LPS-induced mice.

The GCs in the gastrointestinal tract of mammals are characterised by their goblet-shaped morphology and highly polarised columnar epithelial structure [38]. These cells primarily secrete the mucin MUC2, an essential component of the mucosal barrier [39]. The mucosal layer, which is present in the intestinal contents and intestinal epithelial cells, provides lubrication and acts as a buffer barrier [40]. GCs are directly responsible for determining the quality of the mucus [40]. When stained with PAS reagent, GCs exhibit a strong positive reaction, appearing red or purple [41]. As shown in Figure 4 and Appendix A, the GC counts were significantly lower in the duodenum, jejunum and ileum of the LPS group than in those of the control group (*p* < 0.01). Moreover, MPTAs could increase the number of GCs. The effects of MPTAs in the ileum and duodenum were dose-dependent; however, in the jejunum, a high dose of MPTAs exhibited the greatest effect on GCs.

### 3.5. Effects of MPTAs on Serum Cytokine Levels in LPS-Induced Mice

As depicted in Figure 5, the serum levels of TNF-α, IL-8, IL-1β and IL-6 in the LPS group were markedly higher than those in the control group (*p* < 0.01), suggesting a pronounced inflammatory response following intraperitoneal LPS administration. In contrast, the levels of IL-8 and TNF-α in the LPS + Berberine group were significantly lower than those in the LPS group (*p* < 0.05). Furthermore, compared with the LPS group, MPTA treatment significantly reduced the levels of IL-1β and IL-6 (*p* < 0.05). The LPS + H-MPTA group exhibited significantly reduced levels of TNF-α, IL-8, IL-1β and IL-6 (*p* < 0.05); these levels were similar to those noted in the control group. These results indicate the potential of MPTAs in mitigating the LPS-induced inflammatory response in mice.

### 3.6. Effects of MPTAs on Inflammatory Response in LPS-Induced Mice

The potential inhibitory effects of MPTAs on the inflammatory response in LPS-stimulated mice were determined by assessing the relative mRNA levels of IL-1β, NLRP3, TLR4 and NF-κB in the duodenum and colon of the mice. As shown in Figure 6, the mRNA levels of IL-1β, NLRP3, TLR4 and NF-κB in the duodenum and colon of mice in the LPS group were significantly higher than those of mice in the control group (*p* < 0.01). In particular, the mRNA level of TLR4 exhibited a notable increase (*p* < 0.05), indicating that intraperitoneal injection mice of after LPS exhibited severe inflammation.

The mRNA levels of IL-1β, NLRP3 and NF-κB in the LPS + Berberine group were significantly lower than those in the LPS group (*p* < 0.01), indicating a favourable therapeutic effect of Berberine. Furthermore, in both LPS + M-MPTA and LPS + H-MPTA groups, the mRNA levels of IL-1β, NLRP3 and NF-κB were significantly lower than those in the LPS group (*p* < 0.05). These findings suggest that LPS-induced inflammation in mice can be mitigated by MPTAs.

### 3.7. Effects of MPTAs on NF-κB and NLRP3 Pathways

To determine the mechanism by MPTAs can alleviate intestinal inflammation in mice, the effect of MPTAs on the NF-κB and NLRP3 pathways was assessed. The protein levels of IL-18, NLRP3, TLR-4, MD-2, MyD88, NF-κB p65 and NF-κB p-p65 were higher in the LPS group than in the control group. This result suggests that upon stimulation, LPS binds to the TLR4 receptor, thereby activating and upregulating the MyD88 protein and subsequently triggering the activation of the NF-κB pathway.

Upon the stimulation of tissues or cells by LPS, inflammatory responses and oxidative stress reactions are induced, leading to the activation of the NF-κB pathway [42]. Activated NF-κB then triggers the activation of inflammasomes, such as NLRP3, which in turn activate caspase-1 [43]. Therefore, the inhibition of NF-κB transcription can effectively reduce the activation of NLRP3 inflammasomes and restrict pro-inflammatory responses [44]. As illustrated in Figure 7A,B, the protein levels of MD-2 and MyD88 were significantly lower in the LPS + Berberine and MPTA groups than in the LPS group (*p* < 0.05). The protein level of p-p65 in the LPS group was notably higher than that in the control group (*p* < 0.01), suggesting an increase in phosphorylation. The translocation of the p65 protein to the nucleus leads to inflammation. In contrast, the LPS + Berberine and MPTA groups exhibited reduced levels of p65 phosphorylation in a dose-dependent manner. The LPS + M-MPTA and LPS + H-MPTA groups exhibited significantly reduced levels of p-p65 (*p* < 0.01). Moreover, the LPS + H-MPTA group (Figure 7C,D) exhibited significantly reduced levels of both NLRP3 and IL-18 (*p* < 0.01). By impeding the interaction between LPS and the TLR4 receptor, MPTA diminishes the protein expression levels of MD-2 and MyD88, thereby impacting the phosphorylation status of p65. Consequently, the translocation of p65 to the nucleus is hindered, thereby impeding its involvement in anti-inflammatory processes.

In conclusion, the findings suggest that MPTAs can attenuate intestinal inflammation in mice, potentially serving as a therapeutic agent to modulate the expression of key proteins within the NF-κB and NLRP3 pathways.

### 3.8. Effects of MPTAs on Intestinal Microbiota of LPS-Induced Mice

#### OTU Level

The 16S rDNA gene sequences were assessed in all faecal samples collected from the mice. Each sequence was categorised at the 97% similarity threshold using OTU analysis, followed by bioinformatic processing. Analysis of the stool samples of mice from the six groups, namely the control, LPS, LPS + Berberine, LPS + L-MPTA, LPS + M-MPTA and LPS + H-MPTA groups, revealed a total of 1043 OTUs, 12 phyla, 18 classes, 28 orders, 58 families, 162 genera and 292 species. The petal map in Figure 8A presents the number of OTUs in the intestines of mice in different groups. In total, 575 core OTUs were identified in all samples. Nine OTUs were unique to the control group, two were unique to the LPS group, five were unique to the LPS + Berberine group, thirteen were unique to the LPS + L-MPTA group, thirty were unique to the LPS + M-MPTA group and eight were unique to the LPS + H-MPTA group. A sparse curve of each group at the OTU level is presented in Figure 8B. The observed stabilisation of all curves suggested that a substantial proportion of microbial diversity in the samples was captured at this sequencing depth, thereby enhancing the credibility of the results. In the horizontal direction, the curve width reflects the species abundance; the LPS + M-MPTA group was found to exhibit the widest range, followed by the LPS + H-MPTA and control groups, indicating the following order of increase in species abundance: LPS + M-MPTA group, control group and LPS + H-MPTA group.

### 3.9. α-Diversity Analysis

A more detailed examination of the microbial abundance and diversity in the six groups was performed by calculating the ACE, Chao1, Shannon and Simpson indices, in addition to the coverage. As shown in Figure 9, the coverage of the six groups consistently exceeded 0.99, indicating that more than 99% of the flora in all samples was captured. No significant difference was noted among the control, LPS, MPTA and LPS + Berberine groups, indicating that the species diversity and richness in the small intestine of the mice did not change significantly after LPS stimulation.

### 3.10. β-Diversity Analysis

For assessing β-diversity, PCoA and non-metric multi-dimensional scaling (NMDS) were employed to determine the similarity of the microbial community structures among different groups. As shown in Figure 10A, PCoA1 and PCoA2 explained 16.96% and 14.91% of the differences in the microbial community structures among samples, respectively. The LPS and control groups exhibited the greatest disparities in coordinate distances, indicating that LPS treatment can disturb the microbial population structure of the mouse intestine. Furthermore, the microbial compositions between the two groups were significantly divergent. In contrast, the samples from the LPS + H-MPTA group closely resembled those from the control group, indicating that H-MPTA treatment may effectively restore the gut microbial population disrupted by LPS. The results of NMDS analysis revealed that the samples of the control, LPS, LPS + Berberine and MPTA groups could be grouped, with an NMDS analysis of stress, indicating that the key microbial communities in each group had changed (Figure 10B). Furthermore, both analyses demonstrated the most pronounced disparity between the LPS and control groups. On the other hand, the berberine, LPS + L-MPTA and LPS + M-MPTA groups exhibited intermediate positions, suggesting that both therapeutic agents can ameliorate the LPS-induced alterations in the gut microbiota, leading to a shift towards a more normal gut microbiota structure. NMDS analysis revealed that the LPS + H-MPTA group was closest to the control group, indicating that LPS + H-MPTA can regulate the gut microbiota more effectively than berberine.

### 3.11. Effects of MPTAs on Taxonomic Composition of Intestinal Microbiota

The composition of each bacterial group at the phylum level is presented in Figure 11A. The components of the intestinal flora were similar in each group; however, their proportions differed. The important bacterial phyla were Bacteroidota, Firmicutes, Epsilonbacteraeota, Proteobacteria, Actinobacteria and Deferribacterota. Firmicutes species modulate the inflammatory response by facilitating anti-inflammatory mediator secretion [45], while the levels of Proteobacteria and Bacteroides exhibit a positive association with the extent of inflammation. As shown in Appendix A, the Firmicutes/Bacteroidetes proportion in the control, LPS, LPS + Berberine, LPS + L-MPTA, LPS + M-MPTA and LPS + H-MPTA groups was 40.4%, 125.7%, 210.9%, 321.1%, 73.9% and 40.8%, respectively. As shown in Figure 11B, these data indicate that MPTAs can improve the dysbiosis caused by LPS, allowing the microbiome to return to normal levels. The effect in the LPS + H-MPTA group was more significant than that in the LPS + L-MPTA and LPS + M-MPTA groups. MPTAs dose-dependently alleviated the intestinal microbiota composition dysregulated by LPS-induced enteritis in mice at the phylum level. As shown in Figure 11C,D, at the genus level, *norank_f_Muribaculaceae* was dominant in the control group; the decrease in its abundance induced by LPS was alleviated in the LPS + H-MPTA group.

### 3.12. Spearman Correlation Analysis of Serum Cytokines and Gut Microbiota

Heatmap analysis was performed to assess the gut microbiota (Figure 12A). In terms of microbial abundance, the control group exhibited the highest similarity with the LPS + H-MPTA group, while the LPS + L-MPTA and LPS + Berberine groups exhibited the highest similarity with the LPS group. Muribaculaceae, Ruminococcaceae, Lachnospiraceae, Helicobacteraceae and Bacteroidaceae showed a positive correlation with all groups. Peptococcaceae, Erysipelotrichaceae and Clostridiaceae_1 showed a negative correlation with the control and LPS + H-MPTA groups and a positive correlation with the LPS group.

Spearman correlation analysis was performed to assess the correlation between the inflammatory cytokines IL-1β, TNF-α, IL-6 and IL-8 and the 16S rDNA of the gut microbiota. Muribaculaceae and Rikenellaceae were negatively correlated with IL-1β, TNF-α and IL-8, and Prevotellaceae was negatively correlated with IL-1β and IL-8. In contrast, Peptococcaceae, Erysipelotrichaceae and Clostridiaceae_1 were positively correlated with IL-1β, TNF-α and IL-8. In addition, Bacteroidaceae was positively correlated with IL-1β and TNF-α (Figure 12B). These bacterial communities exhibited the highest richness in the control and LPS + H-MPTA groups and the lowest richness in the LPS and berberine groups. Muribaculaceae can regulate immune cells and reduce the levels of pro-inflammatory cytokines [46]. Moreover, Rikenellaceae is associated with the expression levels of chemokine ligand 5 (CCL5), CCL20 and CXCL11 [47]. Prevotellaceae is related to the degradation of hemicellulose and can utilise carbohydrates and proteins to produce succinic acid and acetic acid [48]. Erysipelotrichaceae and Clostridiaceae_1 within the phylum Firmicutes were found to be positively correlated with the inflammatory cytokines IL-1β, TNF-α and IL-8; the richness of these two bacterial communities was the lowest in the control and LPS + H-MPTA groups and highest in the LPS and berberine groups. Research has revealed that Erysipelotrichaceae is associated with colitis [49], while Clostridiaceae_1 can produce butyrate [50]. Therefore, Muribaculaceae, Rikenellaceae and Erysipelotrichaceae may be the key microbial communities in the LPS + H-MPTA group to alleviate gut inflammation. However, this study has certain limitations, as it may not fully encompass the intricate nature of the gut microbiota. Moreover, the selected time points for microbiota analysis may not adequately represent the dynamic microbial changes occurring over time.

### 3.13. Analysis of Volatile SCFAs

The intestinal microbiota produces a multitude of metabolites that are absorbed into the body and circulate through the bloodstream. Many of these metabolites, such as SCFAs, are specifically synthesised by the microbiota. SCFAs have exhibited anti-inflammatory effects in murine models of inflammatory bowel disease (IBD) [51]. As shown in Figure 13, the analysis of volatile SCFAs revealed that LPS-induced mice were mainly affected by changes in acetic acid. The LPS + H-MPTA group exhibited the maximum improvement (*p* < 0.05) in a dose-dependent manner (Appendix A). The proportion of the phylum Bacteroidetes was significantly higher in the LPS + M-MPTA and LPS + H-MPTA groups than in the LPS group; this increase may be related to the increase in acetic acid. Moreover, the proportion of Firmicutes increased in the LPS + L-MPTA and LPS + Berberine groups, which may be related to the increased production of beneficial bacteria. In the LPS + M-MPTA and LPS + H-MPTA groups, inflammatory damage was mainly alleviated by increasing the proportion of Bacteroidetes and promoting the production of SCFAs.

## 4. Conclusions

This study presents novel findings regarding the anti-inflammatory properties of MPTAs in LPS-induced mice. MPTAs effectively reduced intestinal injury and inflammation and modulated gut microbiota homeostasis. The pharmacological activity and underlying anti-inflammatory mechanisms of MPTA were elucidated, suggesting the involvement of the TLR4/NF-κB and NLRP3 signalling pathways (Figure 14). These findings provide strong evidence for the significant anti-inflammatory potential of MPTA, positioning it as a promising candidate for therapeutic intervention in animal gastrointestinal diseases.

## Figures and Tables

**Figure 1 animals-14-02273-f001:**
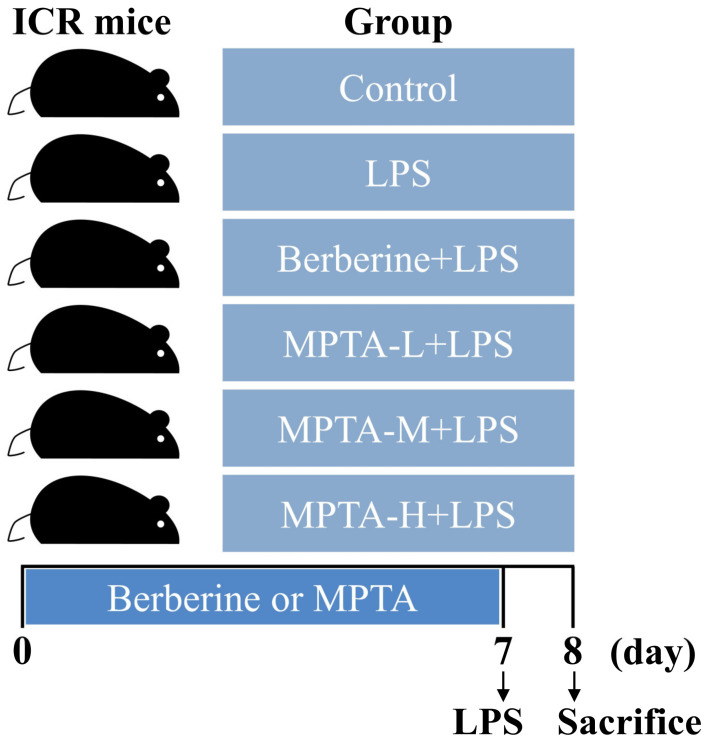
The experimental grouping.

**Figure 2 animals-14-02273-f002:**
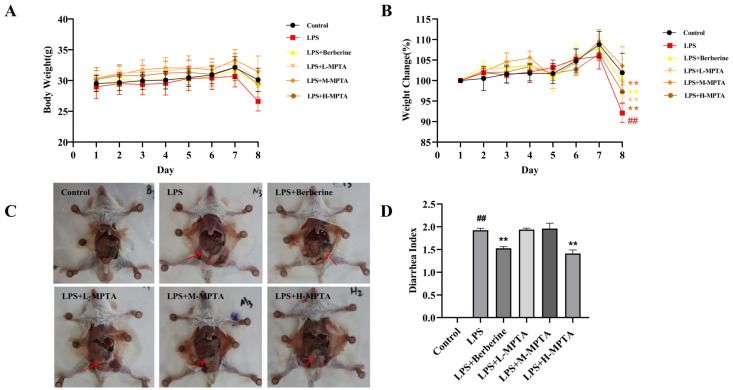
Effects of MPTAs on the symptoms of LPS-induced mice. (**A**) Body weight of LPS-induced mice. (**B**) Weight changes [(final weight − initial weight)/initial weight × 100%] in LPS-induced mice. (**C**) Diarrhoea symptoms in LPS-induced mice. Red arrows indicate that the white intestine and the drained intestinal contents induced by acute diarrhea. (**D**) Diarrhoea index. Results are presented as the mean ± standard error of the mean (SEM). *p*-value < 0.01 (##) versus the control group, and *p*-value < 0.01 (**) versus the LPS group.

**Figure 3 animals-14-02273-f003:**
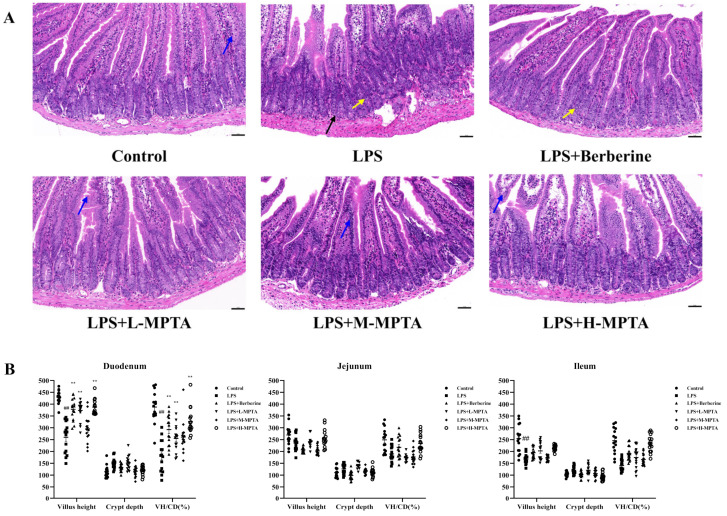
Effects of MPTAs on the duodenal morphology in LPS-induced mice. (**A**) Duodenal morphology based on H&E staining; Intestinal villi shortening, shedding (blue arrow), Villous epithelial cells separate from the lamina propria (black arrow) and inflammatory cell infiltrationin the epithelium of the intestine (yellow arrow); scale bar: 50 μm. (**B**) Statistical analysis of villus height (VH), crypt depth (CD) and VH/CD ratio. Results are presented as the mean ± standard error of the mean (SEM). *p*-value < 0.01 (##) versus the control group, and *p*-value < 0.01 (**) versus the LPS group.

**Figure 4 animals-14-02273-f004:**
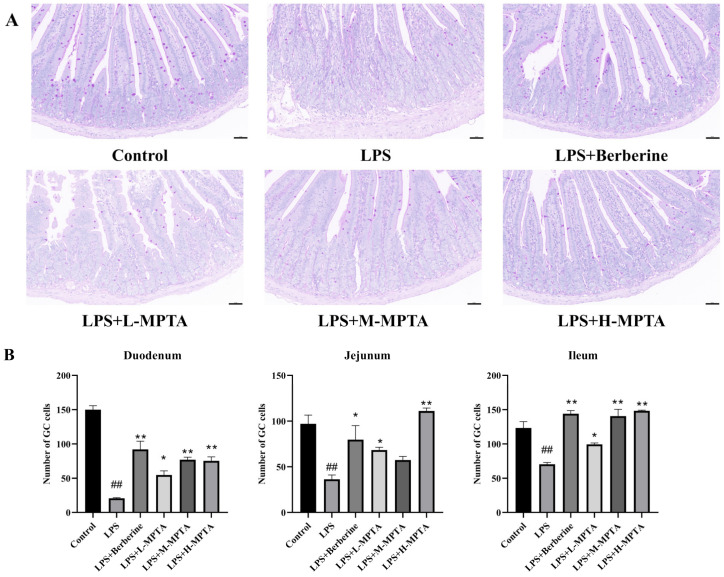
Effects of MPTAs on duodenal goblet cells (GCs) in LPS-induced mice. (**A**) Duodenal GCs assessed using PAS staining; scale bar: 50 μm. (**B**) Statistical analysis of the number of GCs. Results are presented as the mean ± standard error of the mean (SEM). *p*-value < 0.01 (##) versus the control group, and *p*-value < 0.05 (*) or 0.01 (**) versus the LPS group.

**Figure 5 animals-14-02273-f005:**
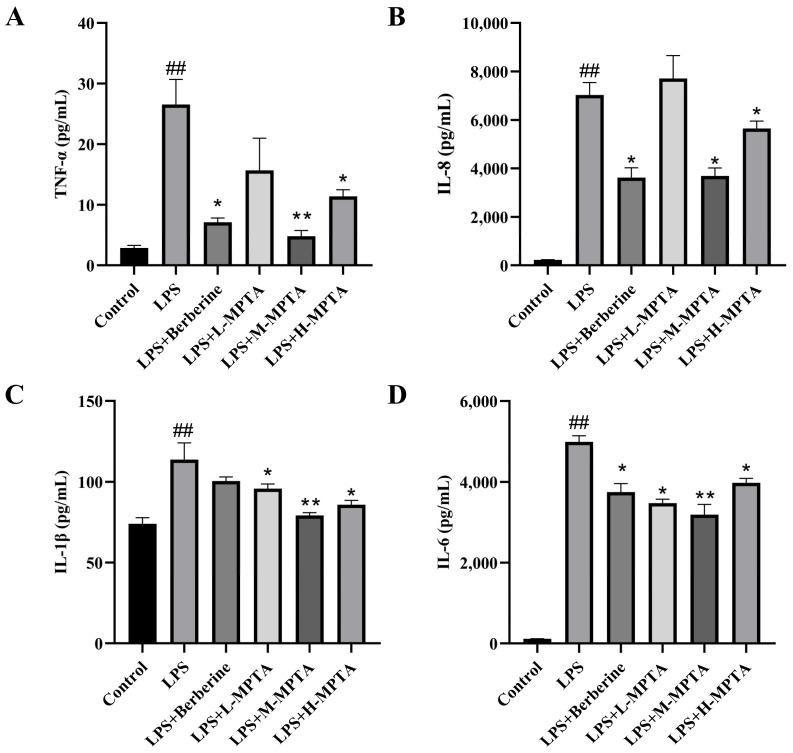
Effects of MPTAs on serum cytokine levels in LPS-induced mice. Assessment of the levels of TNF-α (**A**), IL-8 (**B**), IL-1β (**C**) and IL-6 (**D**) using ELISA kits. Results are presented as the mean ± standard error of the mean (SEM). *p*-value < 0.01 (##) versus the control group, and *p*-value < 0.05 (*) or 0.01 (**) versus the LPS group.

**Figure 6 animals-14-02273-f006:**
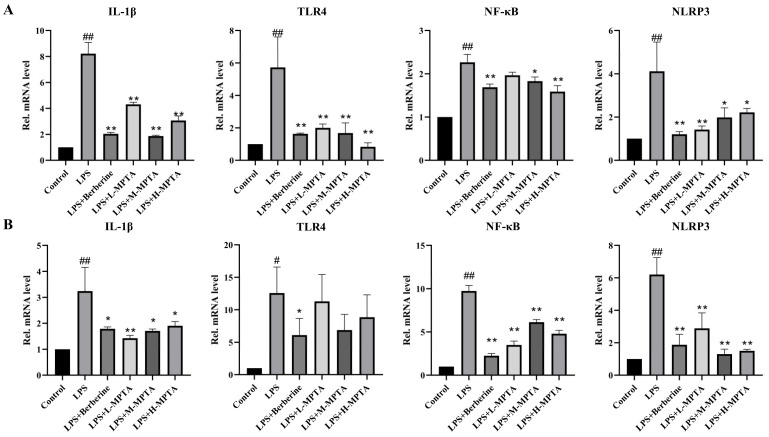
Effects of MPTAs on inflammatory response in LPS-induced mice. Assessment of relative mRNA levels of IL-1β, TLR4, NLRP3 and NF-κB in the duodenum (**A**) and colon (**B**) of mice. Results are presented as the mean ± standard error of the mean (SEM). *p*-value < 0.05 (#) or 0.01 (##) versus the control group, and *p*-value < 0.05 (*) or 0.01 (**) versus the LPS group.

**Figure 7 animals-14-02273-f007:**
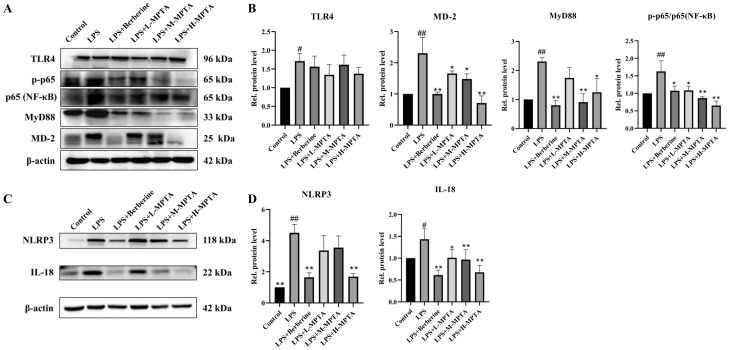
Effects of MPTAs on the NF-κB and NLRP3 pathways. (**A**) Assessment of protein levels of TLR4, MD-2, MyD88, NF-κB p65 and NF-κB p-p65 in the duodenum of mice. (**B**) Statistical analysis of relative expression levels of TLR-4, MD-2, MyD88 compared with internal control protein β-actin and NF-κB p-p65 compared with NF-κB p65. (**C**) Assessment of protein levels of NLPR3 and IL-18 in the duodenum of mice. (**D**) Statistical analysis of relative expression levels of NLPR3 and IL-18 compared with internal control protein β-actin. Results are presented as the mean ± standard error of the mean (SEM). *p*-value < 0.05 (#) or 0.01 (##) versus the control group, and *p*-value < 0.05 (*) or 0.01 (**) versus the LPS group.

**Figure 8 animals-14-02273-f008:**
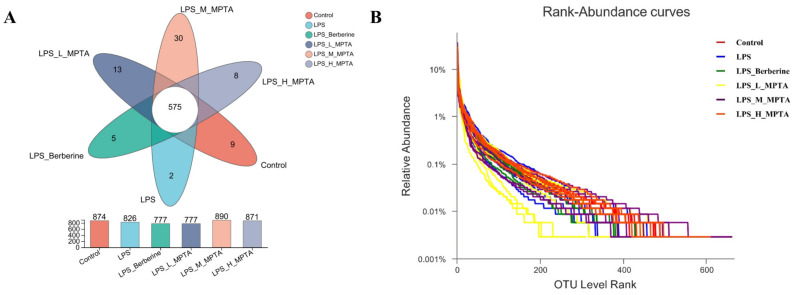
Number of bacterial OTUs in faecal samples of LPS-induced mice. OTU level (**A**) and sparse curve (**B**) of LPS-induced mouse faeces.

**Figure 9 animals-14-02273-f009:**
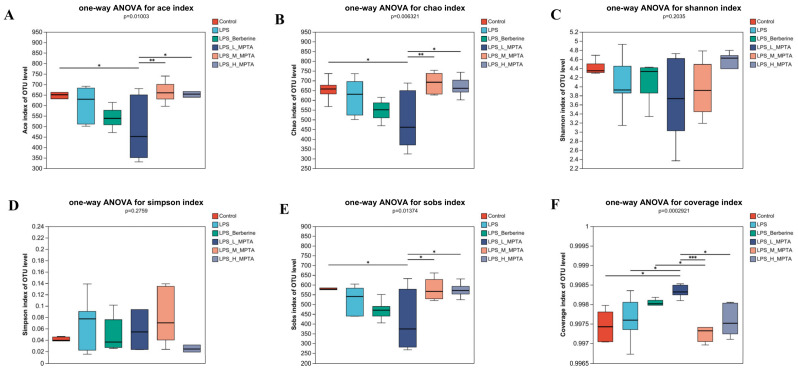
α-diversity analysis of the intestinal microbiota of LPS-induced mice. Assessment of ACE (**A**), Chao1 (**B**), Shannon (**C**), Simpson (**D**) and Sobs (**E**) indices and coverage (**F**) in the faecal samples of LPS-induced mice. Results are presented as the mean ± standard error of the mean (SEM). *p*-value < 0.05 (*), 0.01 (**) or 0.001 (***).

**Figure 10 animals-14-02273-f010:**
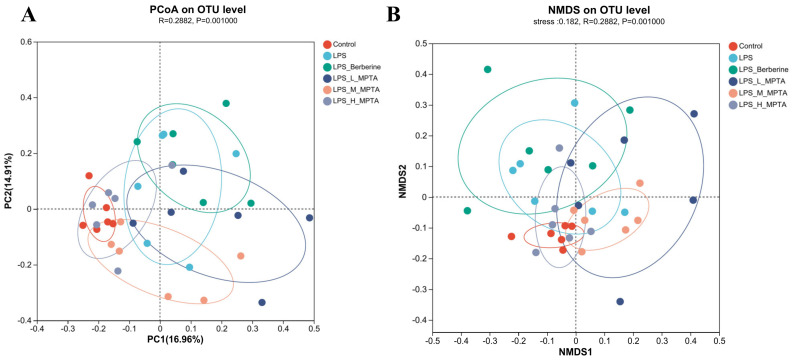
β-diversity analysis of intestinal microbiota of LPS-induced mice. Principal coordinate analysis (PCoA) (**A**) and non-metric multi-dimensional scaling (NMDS) analysis (**B**) of the faecal samples of LPS-induced mice.

**Figure 11 animals-14-02273-f011:**
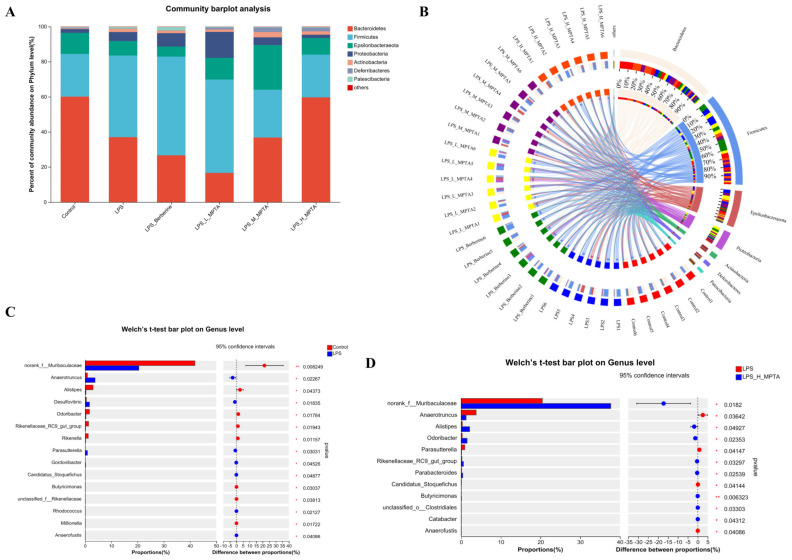
Microbial community composition. (**A**) Composition analysis of each bacterial group at the phylum level. (**B**) Cladogram showing taxa with different abundance in LPS-induced mice. (**C**) Welch’s *t*-test bar plot comparing the microbiome between control and LPS groups at the genus level. (**D**) Welch’s *t*-test bar plot comparing the microbiome between LPS and LPS + H-MPTA groups at the genus level. *p*-value < 0.05 (*) or 0.01 (**).

**Figure 12 animals-14-02273-f012:**
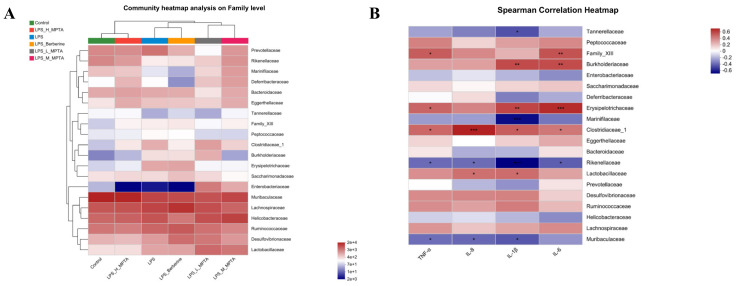
Spearman correlation analysis. (**A**) Heatmap analysis of the gut microbiota. (**B**) Spearman correlation analysis assessing the correlation of inflammatory cytokines with 16 rDNA of the gut microbiota. *p*-value < 0.05 (*), 0.01 (**) or 0.001 (***).

**Figure 13 animals-14-02273-f013:**
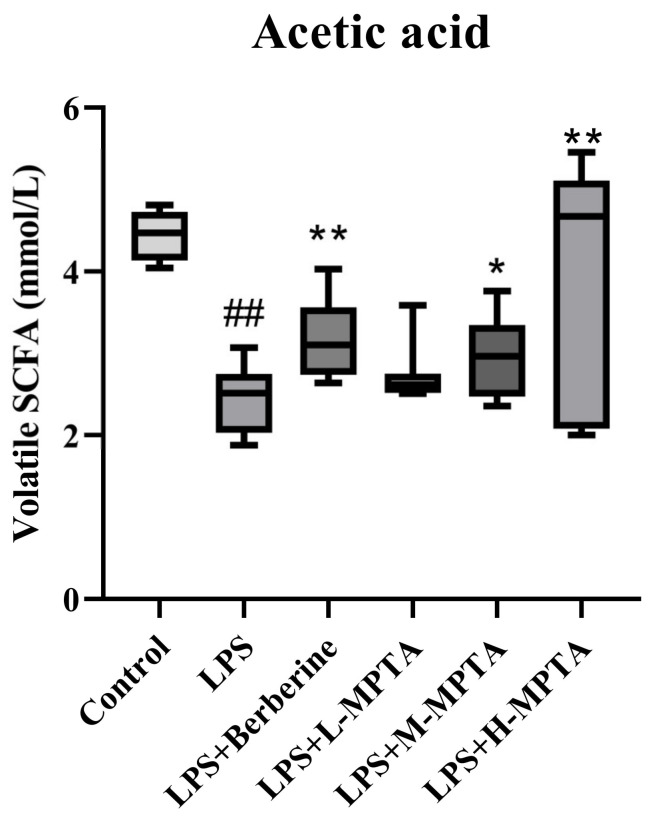
Analysis of volatile acetic acids. Results are presented as the mean ± standard error of the mean (SEM). *p*-value < 0.01 (##) versus the control group, and *p*-value < 0.05 (*) or 0.01 (**) versus the LPS group.

**Figure 14 animals-14-02273-f014:**
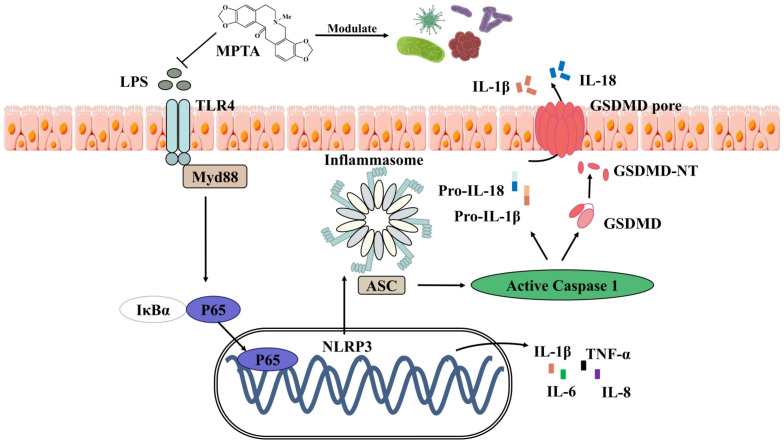
Graphical summary.

**Table 1 animals-14-02273-t001:** Primer sequence.

Gene	Accession Number	Primer Sequence (5′–3′)
*GAPDH*	NM_008084.2	CCTCGTCCCGTAGACAAAATG TGAGGTCAATGAAGGGGTCGT
*TLR4*	NM_021297.2	TGAGGACTGGGTGAGAAATGAGC CTGCCATGTTTGAGCAATCTCAT
*P65*	NM_001365067.1	TCCTTTTCTCAAGCTGATGTGC TTTCGGGTAGGCACAGCAAT
*NLRP3*	NM_145827.4	AGCCTTCCAGGATCCTCTTC CTTGGGCAGCAGTTTCTTTC
*IL-1β*	NM_008361.4	ACTCATTGTGGCTGTGGAGA TTGTTCATCTCGGAGCCTGT

## Data Availability

The data used to support the result of the present study can be obtained from the corresponding authors.

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
