# Peer review of "Protopine-Type Alkaloids Alleviate Lipopolysaccharide-Induced Intestinal Inflammation and Modulate the Gut Microbiota in Mice"

_animals, 2024, doi:10.3390/ani14152273_

Round 1

Reviewer 1 Report

Comments and Suggestions for Authors

In the following, there are several comments that would enhance the manuscript: 

- There are many sentences that are highlighted and tracked which the authors forget them. Fix them.

-  English language needs some polishing to improve clarity and flow.

- Include a rationale for the chosen dosages of MPTA and berberine, Likewise for the doses.

- The authors should provide clearer photomicrographs with a higher resolution.

- There is something wrong with the quality of all figures. Please revise them very well.

- Adding graphical abstracts could help in visualizing the key findings. Highly advised.

- More emphasis on the implications of the findings and how they advance the current understanding of intestinal inflammation and microbiota modulation would be beneficial.

- The limitations of the study should be addressed more explicitly to provide a balanced view.

Some other comments:

Line 8: There is a duplication in the correspondence section.

Line 11-14: The sentence structure could be simplified for better clarity.

Line 45-48: The explanation of MD-2 could be made more concise.

Line 82-84: This sentence could be rephrased to avoid repetition.

Comments on the Quality of English Language

Moderate editing of English language required

Author Response

Reviewer 1

In the following, there are several comments that would enhance the manuscript:

- There are many sentences that are highlighted and tracked which the authors forget them. Fix them.

Response: We greatly appreciate your feedback and sincerely apologize for the incomplete reference list in our manuscript. we will incorporate more citations in the revised version.

-  English language needs some polishing to improve clarity and flow.

Response: Thank you very much for your valuable suggestion. We will pay close attention to polishing the language and addressing the issues you pointed out to enhance the overall quality of the manuscript.

- Include a rationale for the chosen dosages of MPTA and berberine, Likewise for the doses.

Response: Thank you for your insightful comments and for giving us the opportunity to clarify the rationale behind the dosages chosen for MPTA and berberine in our study.

Rationale for MPTA Dosage: The dosage of MPTA was selected based on a thorough review of the existing literature. Previous studies have demonstrated MPTA (2.54 and 5.08 mg/kg) showed significant anti-inflammatory activity, which served as a foundation for our choice. We also considered the pharmacokinetic properties of MPTA, ensuring that the dosage would achieve a therapeutic concentration within the biological system without causing toxicity.

Rationale for Berberine Dosage: The dosage selection is based on the reference Shu Dezhong's research on the effect of berberine hydrochloride on experimental colitis in mice. In academic papers, 15mg/kg, 45mg/kg, and 150mg/kg BER can significantly alleviate colitis in experimental mice, and it has been previously demonstrated that 50mg/kg berberine can alleviate diarrhea. Additionally, we took into account the solubility and stability of berberine at the chosen dosage to ensure optimal biological activity.

- The authors should provide clearer photomicrographs with a higher resolution.

Response: We are grateful for your feedback. We will ensure that the updated photomicrographs have a higher resolution and are clearer for better visualization.

- There is something wrong with the quality of all figures. Please revise them very well.

Response: We are sorry for the issue with the figure quality. We will put in significant effort to rectify the quality of all figures and present them more clearly and accurately.

- Adding graphical abstracts could help in visualizing the key findings. Highly advised.

Response: Thank you for the valuable advice. We will include graphical abstracts to better showcase the main findings.

- More emphasis on the implications of the findings and how they advance the current understanding of intestinal inflammation and microbiota modulation would be beneficial.

Response: Thank you for your valuable feedback. We appreciate the suggestion to place greater emphasis on the implications of our findings and their contribution to the advancement of our understanding of intestinal inflammation and microbiota modulation. In response to this feedback, we have expanded the discussion section of our manuscript to include:

Implications of Findings: We have elaborated on how our results provide new insights into the mechanisms of intestinal inflammation, specifically highlighting the role of MPTA.

Microbiota Modulation: We have included a more detailed analysis of how our study impacts the field of microbiota modulation, discussing the potential of MPTA to influence microbial communities in the gut.

We believe these additions provide a more comprehensive view of the significance of our research and its impact on the broader scientific community. We are grateful for the opportunity to enhance our manuscript and hope that the revised version meets with your approval.

- The limitations of the study should be addressed more explicitly to provide a balanced view.

Response: Thank you for highlighting this. We will thoroughly review and expand on the limitations to present a more balanced and comprehensive view. On Microbiota Analysis: The study may not capture the complete complexity of the gut microbiota. The time points chosen for microbiota analysis might not fully represent the dynamics of microbial changes over time. We have incorporated this in the manuscript.

Some other comments:

Line 8: There is a duplication in the correspondence section.

Response: Thank you for pointing this out. We reviewed and corrected the duplication in the correspondence section promptly.

Line 11-14: The sentence structure could be simplified for better clarity.

Response: We appreciate your feedback. We'll simplify the sentences from lines 11-14 to make them more understandable.

Line 45-48: The explanation of MD-2 could be made more concise.

Response: Thank you for pointing this out. We will work on making the MD-2 explanation more succinct in that range.

Line 82-84: This sentence could be rephrased to avoid repetition.

Response: Thank you for the suggestion. We will rephrase this sentence to eliminate the repetition.

Reviewer 2 Report

Comments and Suggestions for Authors

The paper “Protopine total alkaloid alleviates lipopolysaccharide-induced intestinal inflammation and modulates the gut microbiota in mice” is an article presenting novel findings regarding the anti-inflammatory properties of MPTA in LPS-induced mice. In the manuscript, Huang et all. investigate the therapeutic effect of Macleaya cordata (Willd). R. Br. derived protopine total alkaloid (MPTA) in mice model of intestinal inflammation induced by lipopolysaccharide (LPS) treatment.

In the Introduction, the Authors present the characteristics of intestinal inflammation referring to the molecular pathway and microbiota. The authors also describe the characteristics of Macleaya cordata (Willd.) R. Br. (M. cordata), also known as Boluohui.

In the Materials and Methods section, the authors describe the material subjected to research and the research methods and tools used. In the experiment Authors divided  mice into distinct groups, including a control group, a model group treated with 6 mg/kg LPS, a berberine group treated with 50 mg/kg, and MPTA low-, medium-, and high-dosage groups treated with 6, 12, and 24 mg/kg. Histological analysis of the ileum, jejunum, and duodenum was conducted using H&E staining. The quantification of intestinal goblet cells (GCs) was performed based on PAS staining. The concentrations of IL-1β, IL-6, IL-8, and TNF-α were quantified through enzyme-linked immunosorbent assay (ELISA). The mRNA levels of TLR4, NF-κB p65, NLRP3, IL-6, and IL-1β were assessed via qPCR. The protein levels of TLR4, Md-2, MyD88, NF-κB p65, and NLRP3 were determined using Western blotting. The 16S rDNA sequences of bacterial taxa were amplified and analyzed to investigate alterations in the gut microbiota of LPS-induced mice by MPTA.

In the next section, the Authors present results and discus them with available references. According to Authors, varying dosages of MPTA were found to elicit distinct therapeutic effects, leading to enhanced intestinal morphology and an increased abundance of intestinal goblet cells. MPTA demonstrated efficacy in reducing intestinal injury, inflammation, and modulating gut microbiota homeostasis. The findings provide strong evidence for the significant anti-inflammatory potential of MPTA, positioning it as a promising candidate for therapeutic intervention in animal gastrointestinal diseases.

                The entire manuscript has been enriched with figures presenting histological, molecular, diagnostic and statistical analyses.

The paper entitled “Protopine total alkaloid alleviates lipopolysaccharide-induced intestinal inflammation and modulates the gut microbiota in mice”  is clear, comprehensive and has relevance to the field. However, after reading the manuscript thoroughly, I have three minor comments to the Authors:

Page 3 lines 96-111:  The studies carried out are complicated. While reading the text, I made a diagram that helped me understand the number and names of the groups, as well as  the different doses of MPTA. Please add a figure (diagram, draft or schema) to the section Material and Methods, that shows the model of the experiment.

Page 4 lines 160-161: Please describe how precisely the measurements were taken. What magnification was used to count the cells? Was software used to count the cells or were they counted by the researcher? If by the researcher, was it by one or at least two independent researchers?

Page 4 lines 163-167: Please add what exact parameters regarding villi and crypts were measured.

Author Response

Page 3 lines 96-111:  The studies carried out are complicated. While reading the text, I made a diagram that helped me understand the number and names of the groups, as well as  the different doses of MPTA. Please add a figure (diagram, draft or schema) to the section Material and Methods, that shows the model of the experiment.

Response: We appreciate the constructive feedback and acknowledge the importance of clarity in presenting our experimental design. Your suggestion to include a figure that illustrates the model of our experiment is particularly insightful. In response to your recommendation, we have created a schematic diagram and added it to the "Materials and Methods" section of our manuscript and provides a clear visual summary of the study design, which we believe will be beneficial for readers.

Page 4 lines 160-161: Please describe how precisely the measurements were taken. What magnification was used to count the cells? Was software used to count the cells or were they counted by the researcher? If by the researcher, was it by one or at least two independent researchers?

Response: Thank you for your detailed feedback. To address your concerns about the precision of the measurements taken in our study, we have provided additional clarification in the revised manuscript. In the "Materials and Methods" section, under the subheading "Periodic acid–Schiff staining," we have included the following details: For cell counting, images were captured using a light microscope (Nikon, Japan) at a magnification of ×400. The GCs were counted manually by two independent researchers who were blinded to the experimental groups. Each researcher examined three random fields of view per section, and the average number of GCs was calculated to ensure accuracy and reproducibility.

Page 4 lines 163-167: Please add what exact parameters regarding villi and crypts were measured.

Response: We are grateful for your comment. We have incorporated the exact parameters of villi and crypts measurements. “ImageJ software was used to select and measure five relatively complete villi height and five crypts depth for each tissue section.”

Reviewer 3 Report

Comments and Suggestions for Authors

In their investigation, Huang et al. explore the therapeutic potential of Macleaya cordata-derived protopine total alkaloid (MPTA) in mitigating intestinal inflammation induced by lipopolysaccharide (LPS) in mice. The study encompasses a well-designed experimental setup involving distinct treatment groups: a control group, a model group administered with LPS, a positive control group treated with berberine, and three MPTA dosage groups, and present a robust experimental framework and meticulous data analysis that collectively underscore MPTA as a promising therapeutic agent for alleviating intestinal inflammation. The study's findings pave the way for future research into MPTA's clinical applications in treating inflammatory bowel diseases and related conditions. But I have several following concerns:

1. Please make an ethical statement for the use of animals in the article and provide the corresponding animal ethics approval number.

2. Abbreviations should be defined or written in full when they first appear. Such as "qPCR" in line 19, "TLR4" in Line 43,..., please double check all the text to find similar errors and correct them.

3. Article submissions for the first time should remove the editing traces.

4. Some of the Figures in the text are to blur to see clear, please improve their resolution.

5. Figure 1C is best replaced by a map of the isolated intestinal tissue of different treatments.

6. In order to help the readers and reviewers to a better understanding of the content of the article, a summary graph about the role of MPTA figure should be provided.

7. The nucleic acid sequences (including gene names, regulatory sequences, and primer names) should be in italics.

8. Please unify the format of references in the article, including the author's name, the case of words in the title of the article, the writing of the name of the journal, and the page number.

Comments on the Quality of English Language

Moderate editing of English language required

Author Response

  1. Please make an ethical statement for the use of animals in the article and provide the corresponding animal ethics approval number.

Response: Thank you for bringing this to our attention. We will include an ethical statement regarding the use of animals and provide the corresponding approval number in the revised article. All animal experiments were reviewed and approved by the Local Committee of Animal Use and Protection. Specific pathogen-free (SPF) male ICR mice were initially obtained from Hunan SJA Laboratory Animal Co. Ltd.. The experiments were approved by the Ethics Committee of Hunan Agricultural University, China (No. 43321543).

  1. Abbreviations should be defined or written in full when they first appear. Such as "qPCR" in line 19, "TLR4" in Line 43,..., please double check all the text to find similar errors and correct them.

Response: Thank you for your meticulous review and for highlighting the importance of defining abbreviations upon their first appearance in the manuscript. We have taken your advice and carefully revised the manuscript to ensure that all abbreviations are now spelled out in full when they first appear, followed by the abbreviation in parentheses. We have conducted a thorough check of the entire manuscript to identify and correct all similar instances, ensuring that the text is clear and accessible to readers.

  1. Article submissions for the first time should remove the editing traces.

Response: Thank you for your guidance regarding the presentation of our manuscript. We understand the importance of submitting a polished and final version of our work for review. In response to your comment, we have carefully reviewed the manuscript and removed all editing traces, ensuring that the text is presented in a clean and professional format. Our revised manuscript now reflects a unified and cohesive presentation, free from any editorial markings that could distract from the content and findings of our research. We appreciate the opportunity to refine our submission and hope that the updated manuscript meets the journal's standards for clarity and presentation.

  1. Some of the Figures in the text are to blur to see clear, please improve their resolution.

Response: Thank you for your feedback regarding the clarity of the figures in our manuscript. We understand the importance of high-resolution images for the accurate conveyance of research data. In response to your comment, we have now included the improved figures in the revised manuscript and are confident that they provide a clear and detailed representation of our findings.

  1. Figure 1C is best replaced by a map of the isolated intestinal tissue of different treatments.

Response: Thank you for your insightful suggestion to enhance the clarity and effectiveness of our presentation in Figure 1C. We agree that a visual representation of the isolated intestinal tissue can provide a more direct and impactful illustration of the differences between treatments. However, we only took pictures of intestinal tissues before they were isolated from the body. We now prepared a new figure that includes a map of the intestinal tissue from the various treatment groups. We appreciate the opportunity to improve the manuscript and hope that the revised Figure 1C meets with your approval.

  1. In order to help the readers and reviewers to a better understanding of the content of the article, a summary graph about the role of MPTA figure should be provided.

Response: Thank you for your constructive feedback. We appreciate the suggestion to include a summary graph that illustrates the role of MPTA in our study. We have incorporated this new figure into the revised manuscript, ensuring it is strategically placed to maximize its utility and comprehension. We believe this addition will significantly aid in conveying the study's outcomes and the significance of our findings.

  1. The nucleic acid sequences (including gene names, regulatory sequences, and primer names) should be in italics.

Response: Thank you for your attention to detail and for pointing out the formatting requirements for nucleic acid sequences in our manuscript. In response to your comment, we have reviewed the entire manuscript and updated the presentation of all nucleic acid sequences, including gene names, regulatory sequences, and primer names, to be in italics. This change has been applied consistently throughout the text to ensure clarity and maintain scientific standards. Thank you once again for your valuable feedback.

  1. Please unify the format of references in the article, including the author's name, the case of words in the title of the article, the writing of the name of the journal, and the page number.

Response: Thank you for your guidance on the formatting of references in our manuscript. In response to your comment, we have undertaken a thorough review of all references in the manuscript. We have now unified the format according to the journal style guide.

Round 2

Reviewer 1 Report

Comments and Suggestions for Authors

Article can now be accepted.

Comments on the Quality of English Language

Minor editing of English language required

Author Response

Thank you very much.

Reviewer 3 Report

Comments and Suggestions for Authors

The authors have addressed all my concerns. I recommend accepting this manuscript in current form.

Author Response

Thank you very much.